# Gut Microbiome Dysbiosis and Its Impact on Reproductive Health: Mechanisms and Clinical Applications

**DOI:** 10.3390/metabo15060390

**Published:** 2025-06-11

**Authors:** Efthalia Moustakli, Sofoklis Stavros, Periklis Katopodis, Anastasios Potiris, Peter Drakakis, Stefanos Dafopoulos, Athanasios Zachariou, Konstantinos Dafopoulos, Konstantinos Zikopoulos, Athanasios Zikopoulos

**Affiliations:** 1Laboratory of Medical Genetics, Faculty of Medicine, School of Health Sciences, University of Ioannina, 45110 Ioannina, Greece; katopodisper@gmail.com; 2Department of Informatics and Telecommunications, University of Ioannina, 47100 Arta, Greece; 3Third Department of Obstetrics and Gynecology, University General Hospital “ATTIKON”, Medical School, National and Kapodistrian University of Athens, 12462 Athens, Greece; sfstavrou@med.uoa.gr (S.S.); apotiris@med.uoa.gr (A.P.); pdrakakis@hotmail.com (P.D.); thanzik92@gmail.com (A.Z.); 4Department of Health Sciences, European University Cyprus, Nicosia 2404, Cyprus; stefanosntf2001@gmail.com; 5Department of Urology, School of Medicine, Ioannina University, 45110 Ioannina, Greece; zahariou@otenet.gr; 6IVF Unit, Department of Obstetrics and Gynecology, Faculty of Medicine, School of Health Sciences, University of Thessaly, 41110 Larissa, Greece; kdafop@uth.gr; 7Department of Obstetrics and Gynecology, School of Medicine, Ioannina University, 45110 Ioannina, Greece

**Keywords:** gut microbiota, microbiota–gonadal axis, hormone, immune system, inflammation, fertility

## Abstract

The human gut microbiome is integral to maintaining systemic physiological balance, with accumulating evidence emphasizing its critical role in reproductive health. This review investigates the bidirectional interactions between the gut microbiota and the female reproductive system, mediated by neuroendocrine, immune, and metabolic pathways, constituting the gut–reproductive axis. Dysbiosis, characterized by microbial imbalance, has been linked to reproductive disorders such as polycystic ovary syndrome (PCOS), endometriosis, infertility, impaired spermatogenesis, and pregnancy complications. These associations can be explained by immunological dysregulation, systemic inflammation, altered sex hormone metabolism, and hypothalamic–pituitary–gonadal (HPG) axis disturbances. This review aims to clarify the molecular and cellular mechanisms underpinning gut–reproductive interactions and to evaluate the feasibility of microbiome-targeted therapies as clinical interventions for improving reproductive outcomes.

## 1. Introduction

The billions of microbes that constitute the human gut microbiome significantly influence the host’s endocrine, immunological, and metabolic processes. Recent advancements in microbiome research have elucidated its role as a critical regulator of systemic health, influencing not only the gastrointestinal tract but also other organ systems, including the reproductive system [1]. Bidirectional communication between the gut and reproductive organs is facilitated by a complex network of immunological, metabolic, and neuroendocrine processes known as the gut–reproductive axis (Figure 1) [2].

Mounting evidence suggests that reproductive disorders, including PCOS, endometriosis, infertility, and pregnancy-related complications, may be influenced by gut microbiome dysbiosis, which is characterized by an imbalance in microbial composition and function [3]. Compromised intestinal barrier function, systemic inflammation, immunological dysregulation, and altered estrogen metabolism lead to these outcomes, thereby disrupting reproductive balance. The gut microbiota modulates the metabolism and bioavailability of critical reproductive hormones, impacting the function of the HPG axis [4].

Despite these pathophysiological processes, emerging clinical research is exploring the potential therapeutic benefits of microbiome-targeted treatments, such as dietary modifications, probiotics, prebiotics, and fecal microbiota transplantation (FMT), to improve reproductive outcomes [5]. By restoring microbial balance, these therapies aim to restore immunological and endocrine stability. Understanding how specific microbial taxa, metabolites, and host interactions affect reproductive physiology is crucial for both gaining mechanistic insights and applying this knowledge in reproductive medicine [6].

A comprehensive understanding of the molecular and cellular processes associated with gut microbiota and reproductive health could lead to the development of innovative diagnostic and therapeutic approaches [7]. This review examines current research on the relationship between gut microbial communities and female reproductive health. It investigates the potential therapeutic benefits of microbiome-based therapies in reproductive medicine and clarifies the molecular mechanisms underlying their exchange.

In addition to its role in reproductive health, the gut microbiota is gaining broader recognition as a key factor in sustaining overall physiological balance. This diverse microbial ecology supports several essential functions, including food metabolism, pathogen defense, and immunological control [8]. Dysbiosis, characterized by a disruption of this delicate equilibrium, has been linked to a wide spectrum of diseases, including cardiovascular, metabolic, and neuropsychiatric disorders [9].

Significant focus has been placed on the interplay between gut microbiota and micronutrients, particularly vitamins. Vitamins significantly influence the composition of the gut microbiota, which in turn regulates their absorption and bioavailability. This mutual interaction affects the microbiota’s composition and function, emphasizing the microbiome’s wider significance in human health [10]. This review addresses both domain-specific and general roles of the microbiome as suggested by recent literature by examining the impact of gut microbiota on reproductive health, as well as placing it within the larger framework of systemic health control.

## 2. Materials and Methods

This review was conducted following the Preferred Reporting Items for Systematic Reviews and Meta-Analyses (PRISMA) guidelines to ensure methodological rigor and transparency. A systematic literature search was performed across major scientific databases including PubMed, Scopus, and Web of Science for peer-reviewed articles published between January 2010 and April 2025. The search strategy included combinations of keywords such as “gut microbiota”, “reproductive health”, “dysbiosis”, “infertility”, “PCOS”, “endometriosis”, “microbiome therapy”, and “hormonal modulation”. Articles included were in English and comprised human and animal studies examining the gut–reproductive axis, encompassing peer-reviewed original studies, systematic reviews, and meta-analyses. Studies focusing solely on the vaginal microbiome or unrelated systemic conditions, as well as editorials, commentaries, and case reports, were excluded. The selection process involved an initial screening of titles and abstracts, followed by a full-text review conducted independently by two reviewers, with disagreements resolved by consensus or consultation with a third reviewer. A PRISMA flowchart (Figure 2) summarizes the selection process.

## 3. Mechanistic Links Between the Gut Microbiome and the Reproductive System

The gut microbiota influences reproductive physiology through several interconnected processes, including maintaining immunological tolerance, regulating steroid hormones, and modulating systemic inflammation [11]. Acetate, propionate, butyrate, and other short-chain fatty acids (SCFAs) are microbial metabolites that play a role in energy metabolism and exhibit anti-inflammatory effects. These metabolites impact menstrual regularity and ovarian function by regulating the HPG axis, primarily through their effects on the release of gonadotropin-releasing hormone (GnRH) [12].

Acetate, propionate, and butyrate—key microbial-derived SCFAs—exert their effects on the host through several pathways, notably by binding to G-protein-coupled receptors GPR41 and GPR43 (also known as FFAR3 and FFAR2, respectively) [13]. These receptors are expressed on intestinal epithelial cells, immune cells, and even in hypothalamic tissue. Nuclear factor kappa-light-chain-enhancer of activated B cells (NF-κB), a key regulator of inflammation, can be inhibited through intracellular signaling pathways triggered by SCFA binding [14]. SCFAs reduce systemic inflammation and modulate GnRH secretion at the hypothalamic level by suppressing NF-κB activity. This activity ultimately affects ovarian steroidogenesis and menstrual regularity by influencing the downstream release of follicle-stimulating hormone (FSH) and luteinizing hormone (LH). SCFAs serve as a biological bridge linking the regulation of reproductive hormones with gut microbial activity via integrated metabolic and immune-modulatory mechanisms [15].

In the emerging theory of the gut–brain–reproductive axis, gut microorganisms regulate the production of neurotransmitters like serotonin and γ-aminobutyric acid (GABA), introducing an additional layer of neuroendocrine control over fertility. These neurotransmitters link gut health to neuroendocrine function and the regulation of reproductive hormones by influencing GnRH pulsatility and hypothalamic communication [16,17].

The estrobolome is a collection of microbial genes involved in the metabolism of estrogen and is a crucial hormonal mechanism. The enzyme β-glucuronidase, produced by certain gut bacteria, deconjugates estrogens in the gut, allowing them to be reabsorbed into the bloodstream. Dysbiosis can disrupt this process, leading to insufficient recycling (estrogen deficiency) or excessive reabsorption (hyperestrogenism) [18]. Endometriosis, uterine fibroids, and hormone-sensitive malignancies are examples of estrogen-dependent disorders linked to these abnormalities. Further connecting microbial balance to reproductive outcomes, changing systemic cytokine levels, such as interleukin-6 (IL-6) and tumor necrosis factor-alpha (TNF-α), may also affect endometrial receptivity and ovulatory function [19].

In addition to its effects on immunological and hormonal modulation, the gut microbiota also influences reproductive health through its impact on intestinal barrier integrity. Metabolic endotoxemia is a condition characterized by increased intestinal permeability due to dysbiosis, which allows microbial products like lipopolysaccharides (LPS) to enter the bloodstream [20]. This condition promotes chronic low-grade inflammation, a hallmark of reproductive disorders such as PCOS and unexplained infertility. These inflammatory reactions could disrupt critical reproductive processes like folliculogenesis, implantation, and placental development [4].

Moreover, unprotected sexual intercourse enables the exchange of bacteria between partners, altering the gut and genital microbiota composition and affecting the immune response and infection risk in the reproductive tract. In addition, disruptions in circadian rhythm caused by shift work or irregular sleep patterns have been shown to alter the gut microbiome, which may impact hormone regulation, metabolic balance, and reproductive function [21].

The gut microbiota plays a vital role in regulating female reproductive health through hormonal, immunological, neuroendocrine, and inflammatory pathways, as outlined in Table 1.

## 4. Gut Microbiome Dysbiosis in Reproductive Disorders

There is a growing recognition of the role that gut microbiome alterations play in the pathogenesis of PCOS, a complex endocrine disorder characterized by polycystic ovarian morphology, hyperandrogenism, and ovulatory dysfunction. Research has found that individuals with PCOS have lower microbial diversity and a higher Firmicutes-to-Bacteroidetes ratio, both of which are linked to androgen excess, insulin resistance, and hyperinsulinemia [22]. Disease severity has been correlated with specific microbial shifts, such as an increase in Bacteroides and Escherichia/Shigella, and a decrease in Lactobacillus and Bifidobacterium.

Bacteroides species are plentiful in the human microbiome and can exert both positive and negative effects, depending on the specific strain, its abundance, and the host environment. An overgrowth of certain Bacteroides species has sometimes been linked to dysbiosis and disease symptoms, including inflammation and metabolic abnormalities [23]. However, other Bacteroides strains have also been linked to positive reproductive outcomes, such as a healthy semen profile, most likely via supporting immunological homeostasis and a balanced microbial population. Depending on the context, Bacteroides can act as markers of microbiome health or disease and may also contribute to the development of illness. However, additional research is needed to determine causality and the effects of specific strains [24].

Furthermore, increased gut permeability and a decline in SCFA-producing bacteria could contribute to metabolic dysfunction and systemic inflammation [25]. Recent research has also emphasized the function of microbiota-induced control of interleukin-22 (IL-22), a cytokine essential for preserving glucose homeostasis and the integrity of the intestinal barrier. Reduced IL-22 expression, frequently found in dysbiotic conditions, has been linked to increased intestinal permeability, systemic inflammation, and insulin resistance, key features of PCOS. IL-22 production can be induced by specific beneficial bacteria, particularly those that produce SCFAs, contributing to the protection of metabolic and reproductive functions [26]. Although evidence from various studies indicates a relationship between gut dysbiosis and reproductive disorders, including PCOS and endometriosis, these findings should be approached with careful consideration. The observed microbial changes may be the consequence of common underlying factors like dietary patterns, prescription use (e.g., metformin, antibiotics), or metabolic comorbidities like obesity and insulin resistance rather than a direct causal association. Moreover, the bidirectional nature of the gut–reproductive axis complicates the distinction between primary and secondary alterations in the microbiota [27]. To clarify causality and establish whether dysbiosis contributes to or results from these disorders, additional longitudinal and interventional research is needed.

Through the modulation of immune responses and alterations in estrogen metabolism, gut dysbiosis may play a role in the progression of endometriosis, a chronic inflammatory disorder marked by the ectopic growth of endometrial tissue [28]. Disruptions in the estrobolome, which can lead to increased estrogen dominance, may promote endometrial proliferation and the persistence of lesions. Pro-inflammatory microbial profiles that drive systemic and local inflammatory responses may also aggravate chronic pelvic pain, facilitating the survival of ectopic implants [29,30].

Gut microbial imbalances have also been linked to infertility, recurrent pregnancy loss (RPL), and adverse pregnancy outcomes, including preterm birth, gestational diabetes, and preeclampsia. Immunological dysregulation caused by dysbiosis may impact placental development, implantation, and maternal–fetal tolerance [31]. The maternal gut microbiota during early gestation plays a pivotal role in fetal immune programming and metabolic health. Understanding the microbiome’s role in these reproductive conditions offers novel insights into their pathogenesis and highlights new opportunities for diagnostic and therapeutic intervention [32].

## 5. Microbiota-Driven Modulation of Reproductive Hormones

Through bidirectional communication with endocrine organs and enzymatic modulation of hormone metabolism, the gut microbiota plays a critical role in regulating reproductive hormones. One of the key mechanisms involves microbial β-glucuronidase activity, which deconjugates estrogen metabolites in the intestine, thereby increasing their systemic bioavailability and influencing the HPG axis feedback control [19,33]. Dysbiosis—characterized by reduced microbial diversity or an overabundance of species such as *Bacteroides* that express high levels of β-glucuronidase—can disturb estrogen homeostasis, potentially leading to estrogen dominance or insufficiency. In females, these imbalances may disrupt ovulatory cycles, endometrial receptivity, and embryo implantation [34].

The gut microbiota also modulates androgen metabolism, particularly relevant in conditions such as PCOS. Dysbiosis may enhance androgen biosynthesis through hepatic and intestinal steroidogenic signaling pathways, contributing to hyperandrogenic phenotypes such as hirsutism, acne, and anovulation [35]. Additionally, androgens can in turn influence the microbiota by modulating the host immune response, altering mucosal environments, and affecting microbial metabolism, which shapes the composition and function of microbial communities. This bidirectional interaction highlights how androgen levels and microbiota balance are closely interconnected [36]. Insulin resistance—often driven by gut microbial alterations—amplifies ovarian androgen production, further exacerbating hormonal imbalance [37]. In males, emerging evidence suggests that gut dysbiosis can impair testosterone synthesis, potentially via effects on Leydig cell function, intestinal inflammation, and disruption of lipid and bile acid metabolism, all of which are integral to steroid hormone biosynthesis. Low testosterone is associated with decreased sperm count, motility, and libido [38].

Progesterone, essential for endometrial receptivity, luteal phase support, and pregnancy maintenance, is also influenced by microbial factors [39]. Progesterone resistance refers to a condition in which the endometrial tissue becomes less responsive to progesterone despite normal hormone levels, impairing its ability to prepare the uterus for embryo implantation and sustain early pregnancy [40]. Gut-derived SCFAs and indole compounds play an immunomodulatory role by regulating cytokine profiles and promoting regulatory Treg activity. These actions help establish immune tolerance at the maternal–fetal interface [41,42]. Disruptions in the microbiome may contribute to progesterone resistance, which has been linked to implantation failure, impaired decidualization, and early pregnancy loss.

Moreover, the gut microbiome interacts with the HPA axis, modulating systemic cortisol levels in response to stress. Chronic dysregulation of the gut–brain axis may impair reproductive hormone release by altering GnRH pulsatility, thereby affecting both the HPG and adrenal–gonadal axes. Elevated cortisol levels—induced by microbial imbalance—can suppress gonadotropin secretion, negatively impacting ovulation and spermatogenesis (Table 2) [43].

Together, these endocrine disruptions highlight the gut microbiome’s central role in reproductive homeostasis. Targeting microbial composition and function through dietary interventions, probiotics, or microbiome-derived therapeutics offers promising avenues to restore hormonal balance and enhance fertility outcomes across sexes [44].

## 6. Microbiome–Immune Interactions in Fertility and Pregnancy

Fertility and the establishment of a successful pregnancy depend on both systemic and mucosal immunity, which are significantly influenced by the gut microbiota. Maternal immune tolerance to the semi-allogenic fetus is contingent upon the expansion of regulatory T cells (Tregs) and the production of anti-inflammatory cytokines like IL-10, both of which are promoted by a well-balanced microbial ecosystem [45]. Through the epigenetic control of Foxp3 expression, SCFAs—butyrate in particular—help Treg differentiation. Conversely, gut dysbiosis may lead to aberrant immune activation, increased pro-inflammatory cytokines, and heightened cytotoxic responses, impairing embryo implantation and placental development. Such immunological disruptions have been linked to infertility that cannot be explained and repeated implantation failure [46,47].

The immunological milieu of the reproductive tract is influenced by metabolites and structural elements derived from the microbiota, including LPS and SCFAs, which impact dendritic cell maturation, T cell priming, and antigen presentation [48,49]. Preterm birth, intrauterine growth restriction (IUGR), and preeclampsia are among the adverse outcomes that can arise from disturbances in microbial signals, compromising the immune system essential for maintaining pregnancy. A higher risk of early pregnancy loss and systemic inflammation during gestation has been linked to elevated circulating LPS levels, a defining feature of endotoxemia [50,51].

During pregnancy, dynamic shifts in the maternal gut microbiota correlate with key metabolic adaptations, including increased insulin resistance and lipid metabolism, necessary to support fatal growth [52]. The transgenerational relevance of maternal microbiome health is emphasized by the potential for significant disturbances, such as those associated with obesity-related dysbiosis or gestational diabetes mellitus (GDM), to predispose offspring to chronic metabolic dysfunction [53,54]. These findings indicate that immunomodulatory therapies aimed at the maternal microbiome may offer innovative approaches to enhancing reproductive and perinatal outcomes [55].

The gut–reproductive axis is influenced by external variables that alter the makeup and function of the gut microbiota, including environmental pollutants, antibiotics, and diet. Antibiotic administration during crucial developmental periods may induce sustained microbial dysbiosis, lower short-chain fatty acid production, and impair immunological tolerance. These conditions are associated with poor reproductive outcomes, such as implantation failure and an elevated chance of miscarriage [56]. Dietary factors also play a crucial role; diets rich in fermentable fibers enhance the production of SCFAs and support regulatory immune responses essential for maternal–fetal tolerance, whereas high-fat, low-fiber diets have been shown to reduce microbial diversity and encourage the proliferation of pro-inflammatory taxa [57]. Environmental chemicals such as bisphenol A and phthalates, known for their endocrine-disrupting properties, have been shown to disturb gut microbial balance and weaken the epithelial barrier, which may contribute to systemic inflammation and hormonal disruption. These influences on microbiome composition may contribute to a higher risk of reproductive disorders, particularly in individuals with genetic or metabolic vulnerabilities. The gut–reproductive axis must therefore be seen within a broader ecological framework that accounts for lifestyle decisions and environmental exposures [58].

## 7. Gut Microbiome Dysbiosis and Male Infertility

The gut microbiota interacts with the male reproductive system through the microbiota–gut–testis axis, a complex, bidirectional communication network involving endocrine, immune, and metabolic pathways [49]. Alterations in gut microbial composition can influence testicular function through several mechanisms, including modulation of the HPG axis, immune activation, and the production of microbial metabolites such as SCFAs [59]. Dysbiosis may lead to the overproduction of reactive oxygen species (ROS), resulting in oxidative stress—a major contributor to impaired spermatogenesis and sperm DNA damage. Mechanistically, this occurs through enhanced intestinal permeability (“leaky gut”), which allows lipopolysaccharides (LPS) and other microbial components to enter circulation, triggering systemic inflammation and mitochondrial dysfunction in testicular cells. These effects promote excessive ROS generation, reduce antioxidant defenses such as glutathione, and disrupt redox homeostasis in Sertoli and Leydig cells. Furthermore, gut microbial dysregulation can stimulate pro-inflammatory immune responses, including activation of dendritic cells and macrophages in the testes, leading to chronic local inflammation that disrupts spermatogenic function [60]. Emerging data also suggest that gut microbiota can influence sex hormone levels, thereby impacting reproductive hormone regulation and testicular health.

Animal models provide compelling evidence for the association between gut dysbiosis and male infertility. Ding et al. demonstrated that mice fed a high-fat diet developed gut microbial imbalances characterized by increased Firmicutes and Proteo-bacteria, along with reduced Bacteroidetes and Verrucomicrobia, accompanied by significantly lowered sperm concentration and motility [61]. Similarly, FMT from mice treated with alginate oligosaccharides to mice exposed to busulfan led to the restoration of beneficial gut bacteria—particularly Bacteroidales and Bifidobacteriales—and a notable improvement in sperm parameters. Alginate oligosaccharides were also shown to mitigate spermatogenic inhibition through enrichment of Lactobacillaceae and reduction in pro-inflammatory Desulfovibrionaceae [62]. Additional studies have linked artificial diets and altered gut microbiota to elevated endotoxin levels, epididymal inflammation, and dysregulation of testicular gene expression, all of which contribute to defective spermatogenesis [63]. Moreover, the presence of harmful bacteria such as Staphylococcus aureus has been correlated with abnormal sperm morphology and concentration, while infections with Candida albicans have been shown to impair sperm motility and structure in vitro. These findings underscore the relevance of gut microbiome balance for maintaining optimal male reproductive function and highlight its potential as a target for novel therapeutic strategies, including probiotic supplementation, dietary interventions, and microbiota-directed therapies [64].

## 8. Clinical Implications and Therapeutic Perspectives

Microbiome science holds significant clinical promise for advancing reproductive medicine. Microbiome profiling has the potential to facilitate precision medicine, acting as a non-invasive diagnostic tool for reproductive failure [65]. Distinctive microbial signatures in diseases such as PCOS, endometriosis, and recurrent miscarriages may aid in early diagnosis, prognosis, and the development of personalized treatment plans. As potential indicators of implantation success and pregnancy viability, differences in gut microbial diversity and composition may affect in vitro fertilization (IVF) and other assisted reproductive technologies (ART) [66,67]. A promising strategy for reestablishing microbial balance and enhancing reproductive health is probiotic and prebiotic supplementation. Probiotic and prebiotic supplementation offers a promising strategy for restoring microbial balance and promoting reproductive health [68]. Certain probiotic strains, like Lactobacillus rhamnosus and Bifidobacterium breve, have been shown in clinical studies to improve gut barrier integrity, control endocrine function, and lower systemic inflammation [69]. These effects support ovulatory cycles, endometrial receptivity, and embryo implantation. Furthermore, dietary changes and FMT are being researched as novel approaches to rewire dysbiotic microbiomes and restore immunologic and hormonal equilibrium in reproductive diseases [70].

Despite promising results, strong validation through randomized controlled trials and defined methodologies is necessary for the clinical translation of microbiome-based therapeutics. Challenges include strain-specific effects, inter-individual variability in microbiota composition, and the complexity of host–microbe interactions [71]. Refining treatment techniques will require ongoing study into endocrine–microbial feedback processes, host immunological responses, and microbial metabolites. To fully realize the promise of microbiome regulation in reproductive healthcare, a multidisciplinary approach combining systems biology, reproductive endocrinology, and microbiology will be necessary [72].

Although FMT has demonstrated potential in restoring the balance of gut microbes, its application in reproductive medicine should be carried out carefully, especially in pregnant women or those who are hormone sensitive. Introducing donor microbiota carries risks such as the potential transmission of pathogenic or pro-inflammatory organisms, which could disrupt immune tolerance or hormonal balance, both crucial for sustaining pregnancy [73]. The hypothalamic–pituitary–gonadal axis may be unintentionally modulated, as the long-term endocrine consequences of FMT are not fully understood. Regulatory guidance is still scarce, and few randomized controlled trials currently assess FMT for reproductive health. Until comprehensive safety and efficacy data are established, FMT should be classified as an experimental procedure and implemented under stringent donor selection criteria and meticulous clinical oversight [74].

## 9. Limitations of the Study

Despite its comprehensive scope, it is important to acknowledge several limitations of this evaluation. Notable variability exists across studies in terms of design, participant demographics, microbial assessment techniques, and outcome measures within the current body of literature on the gut–reproductive axis. This heterogeneity restricts the generalizability of conclusions and makes direct comparison of findings more difficult [75].

Additionally, a considerable portion of the existing evidence is derived from cross-sectional studies and preclinical animal models, which provide valuable insights but do not establish causal relationships. The scarcity of comprehensive, long-term randomized controlled trials further limits the ability to draw definitive conclusions about the therapeutic potential of microbiome-based interventions [76].

Reproducibility and the establishment of standardized treatment protocols are further challenged by the lack of microbial strain identification and detailed taxonomic data in numerous studies. It is important to account for potential publication bias, since studies demonstrating beneficial associations between microbiome modulation and reproductive outcomes tend to be published more readily than those with null or negative results [65].

Additionally, although this study primarily focused on the gut microbiota, interactions with other microbiomes such as the vaginal and placental microbiota, which may also have a significant impact on reproductive outcomes, were not thoroughly considered. Ultimately, identifying microbiome influences that are unique to reproductive health is complicated by the intricate interactions between host genetics, dietary habits, environmental exposures, and lifestyle factors that together shape the microbial ecosystem [77].

To enhance the clinical applicability of findings in this emerging field, future research should integrate multi-omic approaches, emphasize methodological consistency, and explore both longitudinal and interventional study designs [78].

## 10. Conclusions

The gut microbiota functions as a fundamental regulator of female reproductive health by influencing immunological, metabolic, and hormonal pathways critical to fertility and pregnancy outcomes. PCOS, endometriosis, and recurrent miscarriages are among the reproductive illnesses that are exacerbated by dysbiosis, which upsets these delicately balanced systems. Mechanistic understanding of the interaction between the gut and the reproductive system has revealed new pathogenic pathways and offered a framework for treatments that target the microbiome to improve reproductive health and restore equilibrium [79].

Developments in computational biology, metabolomics, and high-throughput sequencing have deepened our knowledge of the gut microbiome’s impact on reproductive function as well as its role in systemic physiology [80]. The potential for microbiome-based diagnoses and treatments in reproductive medicine is growing rapidly as our understanding of microbial ecosystems continues to advance. Optimizing fertility, improving pregnancy outcomes, and promoting maternal–fetal health throughout life may all depend on restoring microbial equilibrium through focused therapies [81].

Determining microbial indicators of reproductive health, building standardized methods for microbiome-based therapeutics, and showing causal links between the microbiome and reproductive illnesses should be the goals of future research. The practical application of microbiome-based interventions depends on overcoming inter-individual microbiome variability and developing strain-specific therapies. Applying microbiome science to prevent and treat reproductive disorders can be enhanced through a multimodal approach incorporating host genetics, environmental factors, lifestyle, and nutrition (Figure 3). Reproductive healthcare could be transformed by the implementation of more personalized and efficient therapies, driven by the incorporation of these findings into clinical practice.

## Figures and Tables

**Figure 1 metabolites-15-00390-f001:**
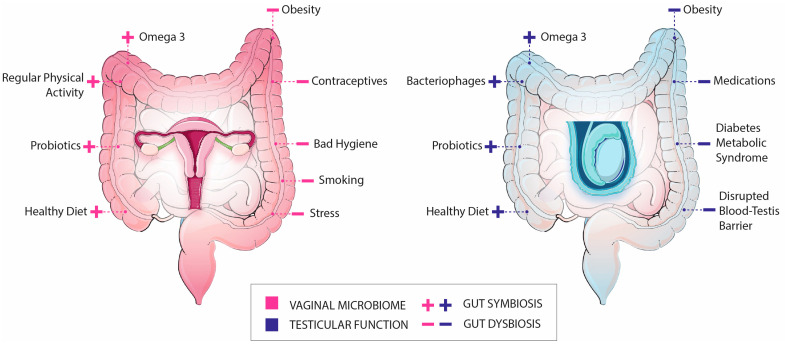
Illustration of factors influencing gut microbiota and their impact on female (vaginal microbiome) and male (testicular function) reproductive health. Positive influences (solid lines) like probiotics, omega-3, and healthy diet promote gut symbiosis, while negative factors (dashed lines) like obesity, stress, and medications contribute to dysbiosis. The image highlights the gut–reproductive axis and its modulation by lifestyle and metabolic conditions.

**Figure 2 metabolites-15-00390-f002:**
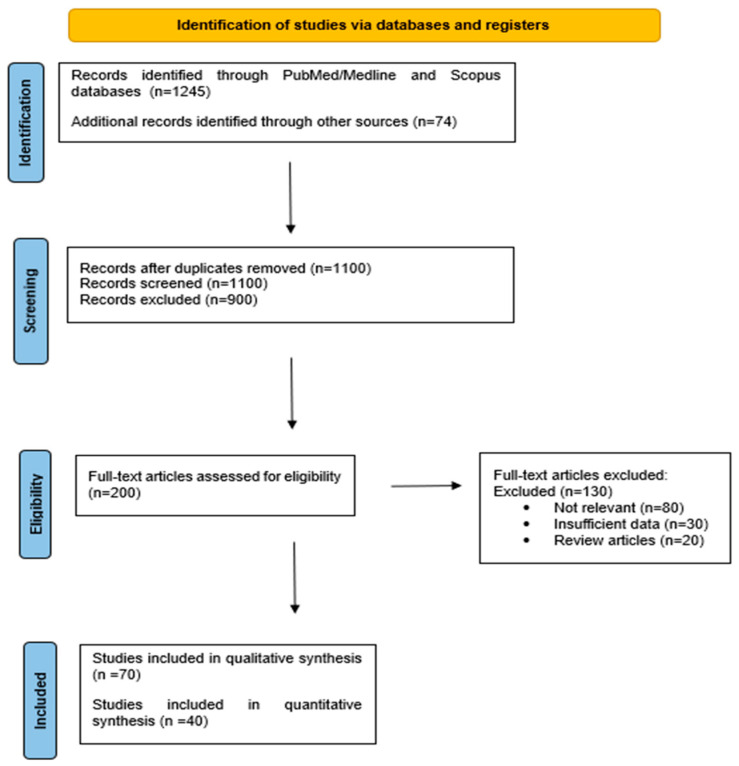
PRISMA flow diagram summarizing the literature selection process.

**Figure 3 metabolites-15-00390-f003:**
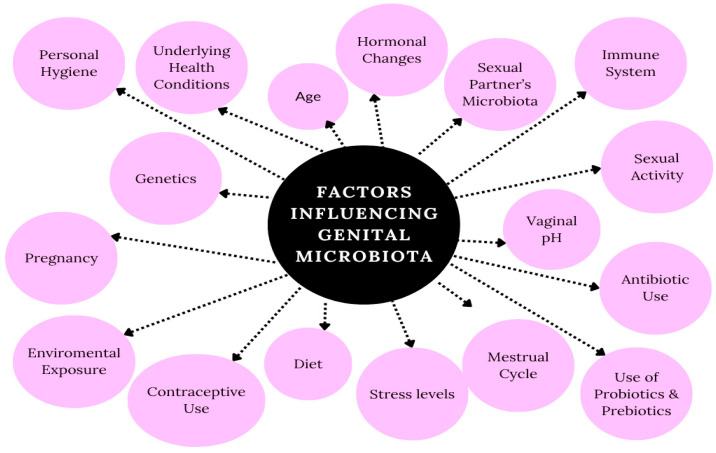
Factors influencing genital microbiota.

**Table 1 metabolites-15-00390-t001:** This table summarizes the primary mechanisms through which the gut microbiome influences reproductive function. It highlights the role of microbial metabolites, hormonal regulation, immune modulation, and intestinal integrity in key reproductive processes.

Mechanism	Microbial Role	Reproductive Impact
Steroid Hormone Regulation	Estrobolome genes metabolize estrogens via β-glucuronidase activity	Estrogen imbalance → endometriosis, fibroids, hormone-dependent cancers
SCFA Production (e.g., butyrate, acetate)	Anti-inflammatory metabolites regulate systemic inflammation	Modulate HPG axis → influence GnRH release, menstrual regularity, ovarian function
Gut–Brain–Reproductive Axis	Microbiota influence neurotransmitters (e.g., serotonin, GABA)	Affects hypothalamic signaling → alters GnRH pulsatility and fertility
Systemic Cytokine Modulation	Control of inflammatory cytokines (e.g., TNF-α, IL-6)	Affects endometrial receptivity, ovulation, implantation
Intestinal Permeability and Endotaxemia	Dysbiosis increases LPS translocation	Induces chronic inflammation → associated with PCOS, infertility

**Table 2 metabolites-15-00390-t002:** Gut microbiota-mediated modulation of reproductive hormones.

Hormone	Microbial Mechanism	Reproductive Impact
Estrogen	β-glucuronidase reactivates conjugated estrogens and affects HPG axis feedback	Females: Estrogen dominance or insufficiency → disrupted ovulation, endometrial dysfunction Males: Altered estrogen:testosterone ratio → impaired spermatogenesis, libido changes
Androgens	Gut dysbiosis enhances androgen biosynthesis via hepatic and intestinal signaling	Females: Hyperandrogenism → PCOS symptoms (acne, hirsutism, anovulation) Males: Altered testosterone levels → reduced sperm quality, testicular dysfunction
Progesterone	SCFAs and indole compounds modulate immune tolerance via Treg cells	Females: Progesterone resistance → implantation failure, pregnancy loss Males: Indirect effect via systemic immune modulation; potential influence on testicular immune privilege
Testosterone	Microbiota composition influences Leydig cell function and steroidogenesis	Males: Low testosterone → hypogonadism, poor sperm production Females: May affect ovarian androgen levels in hyperandrogenic states
Cortisol	Gut–brain axis affects cortisol regulation; microbiota modulate HPA axis stress response	Both: Chronic stress disrupts reproductive hormone balance → menstrual irregularity, reduced sperm quality

## Data Availability

Data sharing is not applicable to this article as no new data were created or analyzed in this study.

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
