# Peer review of "Gut Microbiome Dysbiosis and Its Impact on Reproductive Health: Mechanisms and Clinical Applications"

_metabolites, 2025, doi:10.3390/metabo15060390_

Round 1

Reviewer 1 Report

Comments and Suggestions for Authors

The work addresses a topic of great interest and topicality, namely the role of the gut microbiome in reproductive health, with a broad overview of molecular mechanisms and clinical implications. However, the discussion is often descriptive and lacks an in-depth critical analysis of the evidence, with some statements over-interpreting still preliminary data. Furthermore, the methodology of study selection and evaluation is not sufficiently detailed, limiting the transparency and reproducibility of the review. Overall, the manuscript needs more scientific rigour and clearer organisation to consolidate the added value compared to the existing literature.

- The manuscript is a review, but the header at the top reads “article”. Please correct.

- The introduction is too short. My advice is to describe more about the gut microbiota and its effects on general health. I recommend reading this paper to explore it further: https://doi.org/10.3390/nu17060948

- In general, the text lacks many quotations. For the example a review with an introduction with only 4 quotations does not seem very coherent. There are whole sentences without a single quotation. I recommend adding quotations for all the topics covered.

- The present figure is very beautiful but completely lacks a caption

- Table 2 shows pathologies (e.g. male infertility etc.) that are not described in the text. I advise the authors either to describe these pathologies first or to remove them from table 2, as this is associated with paragraph 3.

- references 21 and 24 are shown in yellow

- Line 151/157-158/174-175/191/214-215/222/234/235 there are dashes in the sentence, please correct.

- Line 214 I consider it unnecessary to describe what dysbiosis of the gut microbiota is since that is what the review is about.

- Line 222 describes that dysbiosis can cause ROS formation without, however, describing any molecular mechanism by which this generation occurs. I advise the authors to further describe and improve this part.

- Line 255 indicate what IVF is.

- Line 260 in which clinical studies? please indicate.

- In general, there is a lack of explanatory pictures that could well describe all the molecular mechanisms underlying the effects described in the text. This would certainly make the review better.

- The method used to search the studies (e.g. databases consulted, keywords, time interval) is not specified. This compromises the reproducibility of the review.

- The authors do not discuss limitations of the cited studies (e.g. sample size, bias in observational studies). Sections analysing contradictions in the literature should be included.

- Associations between dysbiosis and reproductive disorders (PCOS, endometriosis) are presented as causal, without adequate discussion of possible confounding factors (diet, medication, comorbidities).

- The action of SCFAs on the HPG axis is described in a generic manner. Details on specific pathways (e.g. role of GPR41/GPR43 in NF-κB inhibition) are lacking.

- The influence of antibiotics, diet or environmental pollutants in modulating the gut-reproduction axis is not discussed, despite established evidence

- Potential adverse effects of faecal transplantation in pregnancy or on hormonal parameters are not mentioned.

- Introduce a detailed “Materials and Methods” section with PRISMA research strategy

Reviewer 2 Report

Comments and Suggestions for Authors

See the attachment

Round 2

Reviewer 1 Report

Comments and Suggestions for Authors

The authors answered all questions properly.